# Putting Gale & Shapley to Work: Guaranteeing Stability Through Learning

**Hadi Hosseini**
Penn State University, USA
`hadi@psu.edu`

**Sanjukta Roy**
University of Leeds, UK
`s.roy@leeds.ac.uk`

**Duohan Zhang\***
Penn State University, USA
`dqz5235@psu.edu`

## Abstract

Two-sided matching markets describe a large class of problems wherein participants from one side of the market must be matched to those from the other side according to their preferences. In many real-world applications (e.g. content matching or online labor markets), the knowledge about preferences may not be readily available and must be learned, i.e., one side of the market (aka agents) may not know their preferences over the other side (aka arms). Recent research on online settings has focused primarily on welfare optimization aspects (i.e. minimizing the overall regret) while paying little attention to the game-theoretic properties such as the stability of the final matching. In this paper, we exploit the structure of stable solutions to devise algorithms that improve the likelihood of finding stable solutions. We initiate the study of the sample complexity of finding a stable matching, and provide theoretical bounds on the number of samples needed to reach a stable matching with high probability. Finally, our empirical results demonstrate intriguing tradeoffs between stability and optimality of the proposed algorithms, further complementing our theoretical findings.

## 1 Introduction

Two-sided markets provide a framework for a large class of problems that deal with matching two disjoint sets (colloquially agents and arms) according to their preferences. These markets have been extensively studied in the past decades and formed the foundation of matching theory—a prominent subfield of economics that deals with designing markets without money. They have had profound impact on numerous practical applications such as school choice [Abdulkadiroğlu et al., 2005a,b], entry-level labor markets [Roth and Peranson, 1999], and medical residency [Roth, 1984]. The primary objective is to find a *stable* matching between the two sets such that no pair prefers each other to their matched partners.

The advent of digital marketplaces and online systems has given rise to novel applications of two-sided markets such as matching riders and drivers [Banerjee and Johari, 2019], electric vehicle to charging stations [Gerding et al., 2013], and matching freelancers (or flexworkers) to job requester in a gig economy. In contrast to traditional markets that consider preferences to be readily available (e.g. by direct reporting or elicitation), in these new applications preferences may be uncertain or unavailable due to limited access or simply eliciting may not be feasible. Thus, a recent line of work has utilized *bandit learning* to learn preferences by modeling matching problems as *multi-arm bandit* problems where the preferences of agents are unknown while the preferences of arms are known. The goal is to devise learning algorithms such that a matching based on the learned preferences minimize the regret for each agent (see, for example, Liu et al. [2020, 2021], Sankararaman et al. [2021], Basu et al. [2021], Maheshwari et al. [2022], Kong et al. [2022], Zhang et al. [2022], Kong and Li [2023], Wang et al. [2022]).

---

\*Corresponding author

38th Conference on Neural Information Processing Systems (NeurIPS 2024).

| | Uniform agent-DA | Uniform arm-DA | AE arm-DA |
|---|---|---|---|
| Prob. instability | $O(\lvert ES(\overline{m})\rvert \gamma)$ (Thm. 2) | $O(\lvert ES(\underline{m})\rvert \gamma)$ (Thm. 2) | $O\left(\lvert ES(\underline{m})\rvert \exp\left(-\frac{\Delta^2 T_{min}}{8}\right)\right)$ (Thm. 4) |
| Sample complexity | $\tilde{O}(\frac{NK}{\Delta^2}\log(\alpha^{-1}))$ (Thm. 3) | $\tilde{O}(\frac{NK}{\Delta^2}\log(\alpha^{-1}))$ (Thm. 3) | $\tilde{O}(\frac{1}{\Delta^2}\lvert ES(\underline{m})\rvert \log(\alpha^{-1}))$ (Thm. 5) |

Table 1: Comparison of bounds on probability of unstable matchings and the sample complexity to find a stable matching. $\gamma = \exp\left(-\frac{\Delta^2 T}{8K}\right)$.

Despite tremendous success in improving the regret bound in this setting, the study of stability of the final matching has not received sufficient attention. The following stylized example illustrates how an optimal matching (with zero regret) across all agents may remain unstable.

**Example 1.** *Consider three agents $\{a_1, a_2, a_3\}$ and three arms $\{b_1, b_2, b_3\}$. Let us assume true preferences are given as strict linear orderings[1] as follows:*

$$a_1 : b_1^* \succ \underline{b_2} \succ b_3 \qquad\qquad b_1 : \underline{a_2} \succ a_3 \succ a_1^*$$
$$a_2 : b_2^* \succ \underline{b_1} \succ b_3 \qquad\qquad b_2 : \underline{a_1} \succ a_3 \succ a_2^*$$
$$a_3 : b_1 \succ b_2 \succ \underline{b_3^*} \qquad\qquad b_3 : a_1 \succ a_2 \succ \underline{a_3^*}$$

*The underlined matching is the only stable solution in this instance. The matching denoted by $^*$ is a regret minimizing matching: agents $a_1$ and $a_2$ have a negative regret (compared to the stable matching), and $a_3$ has zero regret. However, this matching is not stable because $a_3$ and $b_1$ form a blocking pair. Thus, $a_3$ would deviate from the matching.*

Note that stability is a desirable property that eliminates the incentives for agents to participate in secondary markets, and is the essential predictor of the long-term reliability of many real-world matching markets [Roth, 2002]. Though some work (e.g. Liu et al. [2021], Pokharel and Das [2023]) did stability analysis, it is insufficient as we discuss later in Section 3 and Section 4.

**Our contributions.** We propose bandit-learning algorithms that utilize structural properties of the Deferred Acceptance algorithm (DA)—a seminal algorithm proposed by Gale and Shapley [1962] that has played an essential role in designing stable matching markets. Contrary to previous works [Liu et al., 2020, Kong and Li, 2023], we show that by exploiting an arm-proposing variant of the DA algorithm, the probability of finding a stable matching improves compared to those used in many previous work (an agent-proposing variant, such as Liu et al. [2020], Basu et al. [2021], Kong and Li [2023]). We demonstrate that for a class of profiles (i.e. profiles satisfying $\alpha$-condition or those with a masterlist), the arm-proposing DA is more likely to produce a stable matching compared to the agent-proposing DA for *any* sampling method (Corollary 1). For the commonly studied uniform sampling strategy, we show the probability bounds for two variants of DA for general preference profiles (Theorem 2). We initiate the study of sample complexity in the *Probably Approximately Correct* (PAC) framework. We propose a non-uniform sampling strategy which is based on arm-proposing DA and Action Elimination algorithm (AE) [Even-Dar et al., 2006], and show that it has a lower sample complexity as compared to uniform sampling (Theorem 3 and Theorem 5). Lastly, we validate our theoretical findings using empirical simulations (Section 6).

Table 1 shows the main theoretical results for uniform agent-DA algorithm, uniform arm-DA algorithm, and AE arm-DA algorithm. We note that the novel AE arm-DA algorithm achieves smaller sample complexity for finding a stable matching. Our bounds depend on structure of the stable solution $m$, parameterized by the 'amount' of justified envy $ES(m)$ (see Definition 4.1).

## 1.1 Related Works

The two-sided matching problem is one of the most prominent success story of the field of game theory, and in particular, mechanism design, with a profound practical impact in applications ranging from organ exchange and labor market to modern markets involving allocation of compute, server, or content. The framework was formalized by Gale and Shapley [1962]'s seminal work, where

---

[1]We consider a general case where preferences are given as cardinal values; note that cardinal preferences can induce (possibly weak) ordinal linear orders.

they, along with a long list of subsequent works focused primarily on game-theoretical aspects such as stability and incentives [Roth and Sotomayor, 1992, Roth, 1986, Dubins and Freedman, 1981]. While the DA algorithm is strategyproof for the proposing side [Gale and Shapley, 1962], no stable mechanism can guarantee that agents from *both* sides have incentives to report preferences truthfully in a dominant strategy Nash equilibrium [Roth, 1982]. A series of works focused on strategic aspects of stable matchings [Huang, 2006, Teo et al., 2001, Vaish and Garg, 2017, Hosseini et al., 2021].

Stable matchings under uncertain linear or pairwise preferences were studied by Aziz et al. [2020, 2022]. When preferences are unknown, the problem of learning preferences can be modeled as a multi-agent multi-arm bandit problem. Recent work has shown a variety of learning approaches using Explore-Then-Commit (ETC), Thompson sampling [Kong et al., 2022], in centralized [Liu et al., 2020, Pokharel and Das, 2023] or decentralized [Sankararaman et al., 2021, Kong and Li, 2023] matching markets. Subsequent works focused on domains with restricted preferences (as we also study in this paper) wherein a unique stable matching exists under true preferences [Sankararaman et al., 2021, Basu et al., 2021, Maheshwari et al., 2022, Wang and Li, 2024] or those that generalize to many-to-one markets [Wang et al., 2022, Kong and Li, 2024, Li et al., 2024]. An extensive related work with details on upper bounds is provided in Appendix A.

## 2   Preliminary

Let $[k] = \{1, 2, \ldots, k\}$, and $\zeta(\beta) = \sum_{n=1}^{\infty} \frac{1}{n^\beta}$ denotes the Riemann Zeta function, and $2 > \zeta(\beta) > 1$ if $\beta \geq 2$.

**Problem setup.**   An instance of a two-sided matching market is specified by a set of $N$ agents, $\mathcal{N} = \{a_1, a_2, \ldots, a_N\}$ on one side, and a set of $K$ arms, $\mathcal{K} = \{b_1, b_2, \ldots, b_K\}$, on the other side. The preference of an agent $a_i$, denoted by $\succ_{a_i}$, is a strict total ordering over the arms. Each agent $a_i$ is additionally endowed with a utility $\mu_{i,j}$ over arm $b_j$. Thus, we say an agent $a_i$ prefers arm $b_j$ to $b_k$, i.e. $b_j \succ_{a_i} b_k$, if and only if $\mu_{i,j} > \mu_{i,k}$.[2] We use $\mu$ to indicate the utility profile of all agents, where $\mu = (\mu_{i,j})_{i \in [N], j \in [K]}$. The preferences of arms are denoted by a strict total ordering over the agents, i.e. an arm $b_i$ has preference $\succ_{b_i}$. The *minimum preference gap* is defined as $\Delta = \min_{i \in [N]} \min_{j,k \in [K], j \neq k} |\mu_{i,j} - \mu_{i,k}|$. It captures the difficulty of a learning problem in matching markets, i.e., the mechanism needs more samples to estimate the preference profile if $\Delta$ is small.

**Stable matching.**   A matching is a mapping $m : \mathcal{N} \cup \mathcal{K} \to \mathcal{N} \cup \mathcal{K} \cup \{\emptyset\}$ such that $m(a_i) \in \mathcal{K}$ for all $i \in [N]$, and $m(b_j) \in \mathcal{N} \cup \{\emptyset\}$ for all $j \in [K]$, $m(a_i) = b_j$ if and only if $m(b_j) = a_i$. Additionally, $m(b_j) = \emptyset$ if $b_j$ is not matched. Sometimes we abuse the notation and use $a_i$ to denote $i$ if agent is clear from context, and similarly use $b_j$ to denote $j$ if arm is clear from context. Given a matching $m$, an agent-arm pair $(a_i, b_j)$ is called a blocking pair if they prefer each other than their assigned partners, i.e., $b_j \succ_{a_i} m(a_i)$ and $a_i \succ_{b_j} m(b_j)$. Note that for any arm $b_j$, getting matched is always better than being not matched, i.e., $a_i \succ_{b_j} \emptyset$. A matching is stable if there is no blocking pair.

The *Deferred Acceptance (DA) algorithm* [Gale and Shapley, 1962] finds a stable matching in two sided market as follows: the participants from the proposing side make proposals to the other side according to their preferences. The other side tentatively accepts the most favorable proposals and rejects the rest. The process continues until everyone from the proposing side either holds an accepted proposal (i.e., matched to the one who has accepted its proposal), or has already proposed to everyone on its preference list (i.e., remains unmatched). We consider two variants of the DA algorithm, namely, *agent-proposing* and *arm-proposing*. The matching computed by the DA algorithm is optimal for the proposing side [Gale and Shapley, 1962], i.e. proposing side receives their best match among all stable matchings. It is simultaneously pessimal for the receiving side [McVitie and Wilson, 1971]. We denote the agent-optimal (arm-pessimal) stable matching by $\overline{m}$ and the agent-pessimal (arm-optimal) stable matching by $\underline{m}$.

**Rewards and preferences.**   Agents receive stochastic rewards by pulling arms. If an agent $a_i$ pulls an arm $b_j$, she gets a stochastic reward drawn from a 1-subgaussian distribution[3] with mean

---

[2] We assume preferences do not contain ties for simplicity as in previous works [Liu et al., 2020, Kong and Li, 2023]. When preferences are ordinal and contain ties, stable solutions may not exist in their strong sense (see, e.g. Irving [1994], Manlove [2002]).

[3] A random variable $X$ is $d$-subgaussian if its tail probability satisfies $P(|X| > t) \leq 2\exp(-\frac{t^2}{2d^2})$ for all $t \geq 0$.

value $\mu_{i,j}$. We denote the sample average of agent $a_i$ over arm $b_j$ as $\hat{\mu}_{i,j}$. The *agent-optimal stable regret* is defined as the difference between the expected reward from the agent's most preferred stable match and the expected reward from the arm that the agent is matched to. Formally, we have $\overline{R}_i(m) = \mu_{i,\overline{m}(a_i)} - \mu_{i,m(a_i)}$ for agent $a_i$ and matching $m$. Similarly, the *arm-optimal stable regret* as $\underline{R}_i(m) = \mu_{i,\underline{m}(a_i)} - \mu_{i,m(a_i)}$.

**Preference profiles.** Restriction on preferences has been heavily studied by previous papers (see Sankararaman et al. [2021], Basu et al. [2021], Maheshwari et al. [2022]) as they capture natural structures where, for example, riders all rank drivers according to a common *masterlist*, but drivers may have different preferences according to, e.g., distance to riders. If the true preference profiles are known and there exists a unique stable matching, then the agent-proposing DA algorithm and the arm-proposing DA algorithm lead to the same matching, namely $\overline{m} = \underline{m}$. A natural property of the preference profile that leads to unique stable matching is called uniqueness consistency where not only there exists a unique stable matching, but also any subset of the preference profile that contains the stable partner of each agent/arm in the subset, there exists a unique stable matching. Karpov [2019] provided a necessary and sufficient condition ($\alpha$-condition) to characterize preference profiles that satisfy uniqueness consistency. A preference profile satisfies the $\alpha$-condition if and only if there is a stable matching $m$ and an order of agents and arms such that $\forall i \in [N], \forall j > i, m(a_i) \succ_{a_i} b_j$, and a possibly different order of agents and arms such that $\forall j \in [K], \forall i > j, m(b_j) \succ_{b_j} a_i$.

## 3 Unique Stable Matching: Agents vs. Arms

To warm up, we start by analyzing instances of a matching market where a unique stable solution exists. As we discussed in the preliminaries, these markets are common and can be characterized by a property called uniqueness consistency. We show that for *any* sampling algorithm, the arm-proposing DA algorithm is more likely to generate a stable matching compared to the agent-proposing DA algorithm. All missing proofs and additional results are relegated to the full version Hosseini et al. [2024].

**Theorem 1.** *Assume that the true preferences satisfy uniqueness consistency condition. For any estimated utility $\hat{\mu}$, if the agent-proposing DA algorithm produces a stable matching, then the arm-proposing DA algorithm produces a stable matching.*

Theorem 1 states that when preferences satisfy the uniqueness consistency condition, for any estimated utility, the stability of agent-proposing DA matching implies the stability of arm-proposing DA matching. For any fixed sampling algorithm, each estimation occurs with some probability, so we immediately have the following corollary.

**Corollary 1.** *For any sampling algorithm, the arm-proposing DA algorithm has a higher probability of being stable than the agent-proposing DA algorithm if the true preferences satisfy uniqueness consistency condition.*

The following example further shows that the arm-proposing DA could generate a stable matching even if the estimation is incorrect, while the agent-proposing DA generates an unstable matching. In Section 6, we provide empirical evaluations on stability and regret of variants of the DA algorithm.

**Example 2** (The stability of arm vs. agent proposing DA when estimation is wrong.)**.** *Consider two agents $\{a_1, a_2\}$ and two arms $\{b_1, b_2\}$. Assume the true preferences are as follows:*

$$a_1 : b_1^* \succ \underline{b_2} \qquad\qquad b_1 : a_1^* \succ \underline{a_2}$$
$$a_2 : \underline{b_1} \succ b_2^* \qquad\qquad b_2 : a_2^* \succ \underline{a_1}$$

*The matching denoted by $^*$ is the only stable solution in this instance. Assume that after sampling data, agent $a_1$ has a wrong estimation: $b_2 \succ b_1$ and agent $a_2$ has the correct estimation. Under the wrong estimation, arm-proposing DA algorithm returns the matching $^*$ while agent-proposing DA algorithm returns the underlined matching, which is unstable with respect to true preferences. Note that algorithms that rely on agent-proposing DA (e.g. Liu et al. [2021], Kong and Li [2023]) may similarly fail to find a stable matching as they do not exploit the known arms preferences effectively.*

---
**Algorithm 1:** Uniform sampling algorithm
---
**Input :** Parameter $\beta$, sample budget T.
1 **for** $t = 1, 2, \ldots, T$ **do**
2      **for** $i = 1, 2, \ldots, N$ **do**
3          Agent $a_i$ pulls $m_i(t) = b_j$, where $j = (t + i - 1) \mod K + 1$;
4          Agent $a_i$ observes a return $X_i(t)$ and updates $\hat{\mu}_{i,j}$ ;
5          Compute $UCB_{i,j}$ and $LCB_{i,j}$ using $\beta$;
6      **if** $\exists$ *a permutation* $\sigma_i$ *for all* $i \in [N]$ *such that* $LCB_{i,\sigma_i(k)} > UCB_{i,\sigma_i(k+1)}, \forall k \in [K-1]$ **then**
7          Break;

8 return $\{\hat{\mu}_{i,j} \mid i \in [N], j \in [K]\}$.
---

## 4 Uniform Sampling DA Algorithms

In this section, we compare the stability performance for two types of DA combined with uniform sampling algorithm when the preferences could be arbitrary. Compared with Section 3, we note that the theory in this section does not constrain preferences. We provide probability bounds for finding an unstable matching in Section 4.1 and analyze sample complexity for reaching a stable matching in Section 4.2.

*Uniform sampling* is a technique in bandit literature [Garivier et al., 2016] (usually termed as exploration-then-commit algorithm, ETC). Kong and Li [2023] utilized UCB to construct a confidence interval (CI) for each agent-arm pair, where each agent samples arms uniformly. The exploration phase stops when every agent's CIs for each pair of arms have no overlap, i.e. agents are confident that arms are ordered by the estimation correctly. Then, in the commit phase agents form a matching through agent-proposing DA, and keep pulling the same arms.

Agents construct CIs based on the collected data by utilizing the upper confidence bound (UCB) and lower confidence bound (LCB). Given a parameter $\beta$, if arm $b_j$ is sampled $t$ times by agent $a_i$, we define the UCB and LCB as follows:

$$UCB_{i,j} = \hat{\mu}_{i,j}(t) + \sqrt{2\beta \log(Kt)/t}, \ LCB_{i,j} = \hat{\mu}_{i,j}(t) - \sqrt{2\beta \log(Kt)/t}, \tag{1}$$

where $\hat{\mu}_{i,j}(t)$ is the average of the collected samples.

**Uniform sampling algorithm (Algorithm 1)** Agents explore the arms uniformly. Suppose that agent $a_i$ has disjoint confidence intervals over all arms, i.e., there exists a permutation $\sigma_i := (\sigma_i(1), \sigma_i(2), \ldots, \sigma_i(K))$ over arms such that $LCB_{i,\sigma_i(k)} > UCB_{i,\sigma_i(k+1)}$ for each $k \in [K-1]$. Then, agent $a_i$ can reasonably infer the accuracy of the estimated preference profile. The parameter $\beta$ is used to control the confidence length, where a larger $\beta$ implies that the agent needs more samples to differentiate the utility for a pair of arms.

After the sampling stage, agents can consider to form a matching either through agent-proposing DA or arm-proposing DA, as is discussed in Section 3. For simplicity, we refer uniform sampling ( Algorithm 1) with agent-proposing DA as uniform agent-DA algorithm, and uniform sampling with arm-proposing DA as uniform arm-DA algorithm.

### 4.1 Probability Bounds for Stability

We provide theoretical analysis on probability bounds for stability for the uniform agent-DA algorithm and the uniform arm-DA algorithm. We show probability bounds of learning a stable matching using the properties of stable solutions and the structure of the profile. We first define the following notions of local and global *envy-sets*.

**Definition 4.1.** *The* local envy-set for agent $a_i$ for a matching $m$ *is defined as*

$$ES_i(m) = \begin{cases} \emptyset & \text{if } \{b_j : a_i \succ_{b_j} m(b_j)\} \text{ is empty} \\ \{b_j : a_i \succ_{b_j} m(b_j)\} \bigcup \{m(a_i)\} & \text{otherwise.} \end{cases}$$

*The* global envy-set of a matching $m$ *is defined as the union of local envy-sets over all agents:*

$$ES(m) = \bigcup_{i \in [N]} \{(a_i, b_j) : b_j \in ES_i(m)\}.$$

By the definition of stability, a matching $m$ is stable if and only if the global envy-set is justified, i.e., agents truly prefer their current matched arm to the arms in the envy-set $ES(m)$. Formally, $\mu_{i,m(a_i)} \geq \mu_{i,j}$, for all $(a_i, b_j) \in ES(m)$ if and only if $m$ is a stable matching. This observation is key in establishing theoretical results.

The following lemma provides a condition for finding a stable matching using the envy-set of the estimated matching. The detailed proof of all the results can be found in the full version Hosseini et al. [2024].

**Lemma 1.** *Assume $\hat{\mu}$ is the sample average, and matching $\hat{m}$ is stable with respect to $\hat{\mu}$. Define a 'good' event for agent $a_i$ and arm $b_j$ as $\mathcal{F}_{i,j} = \{|\mu_{i,j} - \hat{\mu}_{i,j}| \leq \Delta/2\}$, and define the intersection of the good events over envy-set as $\mathcal{F}(ES(\hat{m})) = \cap_{(a_i, b_j) \in ES(\hat{m})} \mathcal{F}_{i,j}$. Then if the event $\mathcal{F}(ES(\hat{m}))$ occurs, matching $\hat{m}$ is guaranteed to be stable (with respect to $\mu$).*

Now we prove the following bounds for two types of uniform sampling algorithms. The probability bound depends on the size of the envy-set.

**Theorem 2.** *By uniform sampling (Algorithm 1), each agent samples each arm $T/K$ times, and $\hat{m}_1$ and $\hat{m}_2$ are matchings generated by agent-proposing DA and arm-proposing DA, respectively. Then,*

*(i) $P(\hat{m}_1 \text{ is unstable}) = O(|ES(\overline{m})| \exp(-\frac{\Delta^2 T}{8K}))$,*

*(ii) $P(\hat{m}_2 \text{ is unstable}) = O(|ES(\underline{m})| \exp(-\frac{\Delta^2 T}{8K}))$,*

*where $\underline{m}$ is the agent-pessimal stable matching and $\overline{m}$ is the agent-optimal stable matching.*

*Proof sketch.* Since $\hat{m}_1$ and $\hat{m}_2$ are produced by DA based on $\hat{\mu}$, both matchings are stable with respect to $\hat{\mu}$. By Lemma 1, both matchings are guaranteed to be stable with respect to $\mu$ conditioned on $\mathcal{F}(ES(\hat{m}_1))$ (or $\mathcal{F}(ES(\hat{m}_2))$). Thus, it follows

$$P(\hat{m}_1 \text{ is unstable}) \leq 1 - P(\mathcal{F}(ES(\hat{m}_1)))$$
$$= \mathbb{E}[\cup_{(a_i, b_j) \in ES(\hat{m}_1)} P(|\mu_{i,j} - \hat{\mu}_{i,j}| \geq \Delta/2)] \quad \text{[definition of } \mathcal{F}\text{]}$$
$$\leq \mathbb{E}[\sum_{(a_i, b_j) \in ES(\hat{m}_1)} P(|\mu_{i,j} - \hat{\mu}_{i,j}| \geq \Delta/2)] \quad \text{[union bound]}$$
$$\leq 2\mathbb{E}[|ES(\hat{m}_1)|] \exp(-\frac{\Delta^2 T}{8K}) \quad [\sqrt{\frac{K}{T}}\text{-subgaussian with 0 mean].}$$

To complete the proof for $\hat{m}_1$, we demonstrate an upper bound of $\mathbb{E}[|ES(\hat{m}_1)|]$. We show that $|\mathbb{E}[|ES(\hat{m}_1)|] - |ES(\overline{m})|| \leq N^2 K^3 exp(-\frac{\Delta^2 T}{4K})$. The difference is negligible when $T$ is sufficiently large. Thus the first statement follows. The complete proof is deferred to Appendix C. $\square$

In the next lemma, we show the relation between the size of the envy-sets for the agent-optimal and agent-pessimal matchings. Then, combining Theorem 2 and Lemma 2, we prove the next corollary.

**Lemma 2.** *Given any instance of a matching problem, we have the following relationship between the size of the two envy sets: $|ES(\underline{m})| \leq |ES(\overline{m})|$.*

*Proof.* Agent-pessimal stable matching $\underline{m}$ is the arm-optimal stable matching, and agent-optimal stable matching $\overline{m}$ is the arm-pessimal stable matching, so we have that $\underline{m}(b_j) \succ_{b_j} \overline{m}(b_j)$ or $\overline{m}(b_j) = \underline{m}(b_j)$ for each arm $b_j$. From the definition of the envy-set, we have $|ES(\underline{m})| \leq |ES(\overline{m})|$. $\square$

**Corollary 2.** *The uniform arm-DA algorithm has a smaller probability bound of being unstable than the uniform agent-DA algorithm.*

**Remark 1.** *Liu et al. [2020] showed the probability bound of $\exp(-\frac{\Delta^2 T}{2K})$ for finding an invalid ranking by the ETC algorithm, where a valid ranking is defined as the estimated ranking such that the estimated pairwise comparison is correct for a subset of agent-arm pairs. However, their result did not relate the probability bound with the structure of the instance, whereas the bound in Theorem 2 crucially uses the envy set to improve the probability of finding a stable solution. Liu et al. [2021] provided an upper bound on the sum of the probabilities of being unstable for the Conflict-Avoiding UCB algorithm (CA-UCB). Under CA-UCB algorithm, $O(\log^2(T))$ out of $T$ matchings are unstable in expectation.*

---

**Algorithm 2:** Arm elimination algorithm

---

**Input :** agent $a$, arms $b_1, b_2$, sample sizes $n_1, n_2$, and sample mean $\hat{\mu}_1, \hat{\mu}_2$.

1 Calculate $LCB_1, LCB_2, UCB_1, UCB_2$;
2 $winner \leftarrow$ empty;
3 **while** $\max(LCB_1, LCB_2) < \min(UCB_1, UCB_2)$ **do**
4      $index \leftarrow$ which.min$(n_1, n_2)$;
5      Agent $a$ pulls $b_{index}$;
6      Agent $a$ observes a return and updates $\hat{\mu}_{index}, n_{index}$;
7      Update all $UCB$ and $LCB$;
8 $winner \leftarrow$ index of maximum $(\hat{\mu}_1, \hat{\mu}_2)$;
9 **return** $winner$.

---

## 4.2 Sample Complexity

We turn to analyze the sample complexity to learn a stable matching under the probably approximately correct (PAC) framework. In particular we ask: given a probability budget $\alpha$, how many samples $T$ are needed to find a stable matching? Formally, an algorithm has sample complexity $T$ with probability budget $\alpha$ if with probability at least $1 - \alpha$, the algorithm guarantees that it would find a stable matching with the total number of samples over all agent-arm pairs upper bounded by $T$.

**Theorem 3.** *[Sample complexity for uniform sampling algorithm] With probability at least $1 - \alpha$, both the uniform agent-DA and the uniform arm-DA algorithms find a stable matching with the same sample complexity $\tilde{O}(\frac{NK}{\Delta^2} \log(\alpha^{-1}))$[4].*

Note that uniform agent-DA algorithm finds the stable matching $\overline{m}$, and uniform arm-DA algorithm finds the stable matching $\underline{m}$. Uniform sampling (Algorithm 1) suffers from sub-optimal sample complexity for finding stable matchings since agents sample each arm uniformly. Thus, in the next section we devise an exploration algorithm that exploits the structure of stable matchings by utilizing arms' known preferences.

## 5 An Arm Elimination DA Algorithm

The proposed algorithm (Algorithm 3) combines the arm-proposing DA and Action Elimination (AE) algorithm [Audibert and Bubeck, 2010, Even-Dar et al., 2006, Jamieson and Nowak, 2014]. The AE algorithm eliminates an arm (i.e. no longer sampling the arm) when confidence bound indicates that the arm is sub-optimal (i.e. the upper confidence bound is smaller than another arm's lower confidence bound), and outputs the best arm when there is only one arm that hasn't been eliminated. Note that Algorithm 3 differs from the vanilla arm-proposing DA in Line 8, when an agent has been proposed by two arms. Agents utilize the arm elimination algorithm (see Algorithm 2) until the agent eliminates the sub-optimal arm. Note that at every round, each agent chooses an arm with fewer samples thus far (see Line 4 in Algorithm 2). One significant observation is that if Algorithm 2 outputs winners correctly whenever an agent is proposed, Algorithm 3 terminates with the arm-optimal matching $\underline{m}$.

### 5.1 Probability Bounds for Stability

We compute the probability bound for learning an unstable matching for Algorithm 3. Contrary to uniform sampling, here we compute the bound on given sample size.

**Theorem 4.** *By Algorithm 3, assume that agent $a_i$ samples arm $b_j$ for $T_{i,j}$ and $\hat{m}$ is returned by the algorithm. We define $T_{min} = \min_{(a_i, b_j) \in ES(\hat{m})} T_{i,j}$ as the minimum sample size for agent-arm pairs. Then, we have*

$$P(\hat{m} \text{ is unstable}) \leq O(|ES(\underline{m})| \exp(-\frac{\Delta^2 T_{min}}{8})).$$

**Remark 2.** *Theorem 4 provides stability bound for Algorithm 3 that depends on $T_{i,j}$, which is unknown apriori. If the total sample budget is $NT$ and we set $T_{i,j} = \frac{NT}{|ES(\underline{m})|}$, the stability bound*

---

[4] $\tilde{O}$ denotes the upper bound that omits terms logarithmic in the input.

---

**Algorithm 3:** AE arm-DA algorithm

---
**Input :** arms' preference lists, sample budget $T$.

1   $m \leftarrow$ empty matching;
2   **while** $\exists$ *an unmatched arm b who has not proposed to every agent and sample number* $\leq T$ **do**
3      $a \leftarrow$ highest-ranked agent to whom $b$ has not yet proposed;
4      **if** *a is unmatched* **then**
5         Add $(a, b)$ to $S$;
6      **else**
7         $b' \leftarrow$ arm currently matched to $a$;
8         $a$ finds the preferred arm from $\{b, b'\}$ using arm elimination, Algorithm 2;
9         **if** *a prefers b to b'* **then**
10           Replace $(a, b')$ in $m$ with $(a, b)$;
11           Mark $b'$ as unmatched;

12   Arbitrarily match remaining agents and arms if $m$ is incomplete;
13   **return** $m$

---

becomes $O(|ES(\underline{m})|exp(-\frac{\Delta^2 NT}{8|ES(\underline{m})|}))$, *which is smaller than uniform arm-DA's stability bound* $O(|ES(\underline{m})|exp(-\frac{\Delta^2 T}{8K}))$, *as stated in Theorem 2. Even though the upper bound could be larger than that of Algorithm 1, simulated experiments show that AE arm-DA significantly improves stability guarantees compared to the uniform sampling variants (Algorithm 1).*

### 5.2 Sample Complexity

We compute the sample complexity to learn a stable matching for Algorithm 3. Note that agents only sample pairs in the envy-set, while in Algorithm 1 agents explore all arms uniformly. The following analysis shows that Algorithm 3 has smaller sample complexity compared to Algorithm 1.

**Theorem 5.** *[Sample complexity for AE arm-DA algorithm] With probability at least* $1 - \alpha$, *Algorithm 3 terminates and returns a stable matching,* $\underline{m}$, *with sample complexity of*

$$\tilde{O}(\frac{ES(\underline{m})}{\Delta^2} \log(\alpha^{-1})).$$

*Proof sketch.* We begin by defining a good event $|\hat{\mu}_{i,j}(t) - \mu_{i,j}| \leq \sqrt{2\beta \log(Kt)/t}$ only for agent-arm pairs in the envy-set $|ES(\underline{m})|$. Conditioned on such events for all time, we demonstrate that the algorithm terminates with true preferences on the envy-set $ES(\underline{m})$, and thus, the algorithm executes the arm-proposing DA when agents have known preferences and produces $\underline{m}$.

Then we show the upper bound of sample complexity for each agent-arm pair: $T = O(\frac{\beta}{\Delta^2} \log(\frac{\beta K}{\Delta^2}))$. We prove it by induction on the number of proposals. The base case is when arms $b_j$ and $b_{j'}$ propose to agent $a_i$ for the first time. Then, we show the number of samples for each pair is bounded by $T$. In the inductive step, say $b_j$ is the winner in the last round and has sampled $t_{i,j} \leq T$ times, and $b_{j'}$ proposes to $a_i$ in this round. Then if $a_i$ samples $b_j$ for $T - t_{i,j}$ more times and $a_i$ samples $b_{j'}$ for $T$ times, by the same computation as the base case, we have that the number of samples for each pair is bounded by $T$. Since Algorithm 3 only samples the agent-arm pairs in the envy set $ES(\underline{m})$, we get that the total sample complexity is $|ES(\underline{m})|T = O(\frac{\beta(|ES(\underline{m})|)}{\Delta^2} \log(\frac{\beta K}{\Delta^2}))$. By setting the probability budget $\alpha = \frac{4|ES(\underline{m})|}{K^\beta}$, we have that with probability at least $1 - \alpha$, the AE arm-DA algorithm has sample complexity $\tilde{O}(\frac{|ES(\underline{m})|}{\Delta^2} \log(\alpha^{-1}))$. The complete proof appears in Appendix D. $\square$

It is worth noting that a large $\beta$ implies a small $\alpha$, which implies that the algorithm needs more samples to guarantee finding a stable matching. We show bounds on the envy-sets in the next lemma.

**Lemma 3.** *Considering any true preference* $\mu$, *we have the following bounds for envy-set:*

   *(i) Size of the envy-set for* $\overline{m}$: $(max\{N, K\} - N)N \leq |ES(\overline{m})| \leq NK$.

   *(ii) Size of the envy-set for* $\underline{m}$: $(max\{N, K\} - N)N \leq |ES(\underline{m})| \leq NK - N + 1$.

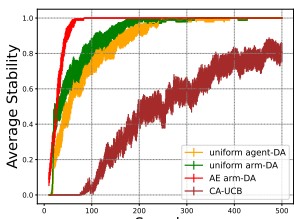 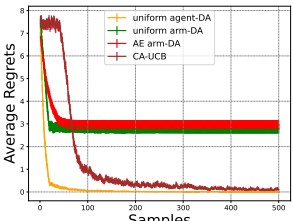 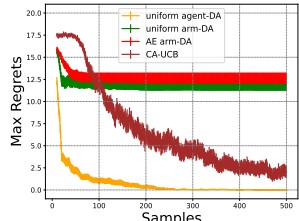

Figure 1: 95% confidence interval of stability and regret for 200 randomized general preference profiles.

**Remark 3.** *By comparing Theorem 5 to Theorem 3, the sample complexity ratio between the AE arm-DA (Algorithm 3) and uniform arm selection (Algorithm 1) is $\frac{|ES(\underline{m})|}{NK}$, which further shows that fewer arms from the envy-set $ES(\underline{m})$ need to be sampled. Lemma 3 states the best-case and worse-case ratios as $\frac{max\{N,K\}-N}{K}$ and $1 - \frac{N-1}{NK} < 1$. Thus, Algorithm 3 strictly improves the sample complexity of finding a stable matching.*

**Remark 4.** *One can illustrate the magnitude of $|ES(\underline{m})|$ through the lens of arm-proposing DA algorithm. Observe that $|ES(\underline{m})|$ is the number of proposals made by arms and rejections made by agents in the arm-proposing DA algorithm. In a highly competitive environment for arms, e.g. when there are much more arms than agents so that many arms are not matched, the magnitude of $|ES(\underline{m})|$ is large. In a less competitive environment, e.g. when arms put different agents as their top choices, $|ES(\underline{m})|$ has much smaller magnitude.*

## 6 Experimental Results

In this section, we experimentally validate our theoretical results. For this, we consider $N = K = 20$ and randomly generate preferences. In particular, we follow a similar experiment setting in Liu et al. [2021]: for each $i$, the true utilities $\{\mu_{i,1}, \mu_{i,2}, \ldots, \mu_{i,20}\}$ are randomized permutations of the sequence $\{1, 2, \ldots, 20\}$ so that the minimum preference gap is fixed ($\Delta = 1$) and algorithm performance exhibits relatively low variability. Arms' preferences are generated the same way. We conduct 200 independent simulations, with each simulation featuring a randomized true preference profile. We compare average stability, i.e., the proportion of stable matchings over 200 experiments, average regrets, and maximum regrets over agents between four algorithms: uniform agent-DA[5], uniform arm-DA, AE arm-DA, and CA-UCB [Liu et al., 2021].

In terms of stability, our experiments show that the AE arm-DA algorithm significantly enhances the likelihood of achieving stability compared to both types of uniform sampling algorithm and CA-UCB algorithm (Figure 1). On the other hand, the regret gap between uniform agent-DA and other two arm-proposing types of algorithms illustrates the utility difference of agent-optimal matching $\overline{m}$ and arm-optimal matching $\underline{m}$. We note that when preferences are restricted to have unique stable matching, AE arm-DA algorithm's regret converges faster to 0, compared to uniform algorithms, while still keeping faster stability convergence (Figure 2). Additional experiments with other preference domains (e.g. masterlist) in provided in Appendix E.

At the first glance, Figure 1 (the center and the right plots) seems to suggest that the the uniform arm-DA is outperforming the AE arm-DA algorithm. However, note that the regret here is with respect to the agent-optimal solution (i.e. $\overline{R}$); and thus, the AE arm-DA algorithm by design is not optimized to reach that solution. Upon further investigation, however, we see that when comparing the two algorithms using the agent-pessimal regret ($\underline{R}$) then the AE arm-DA converges with fewer samples both in terms of average and maximum regrets, as illustrated in Figure 3.

## 7 Conclusion and Future Work

The game-theoretical properties such as stability in two-sided matching problems are critical indicators of success and sustenance of matching markets; without stability agents may 'scramble' to participate

---

[5]The uniform agent-DA algorithm is ETGS algorithm in Kong and Li [2023] with minor differences.

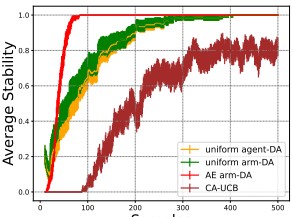 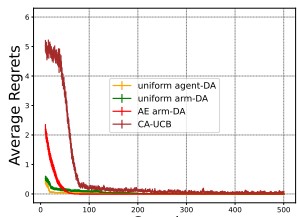 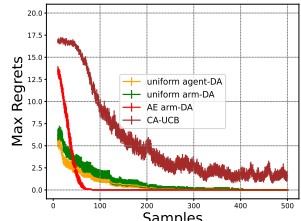

Figure 2: 95% confidence interval of stability and regrets for 200 randomized SPC preference profiles. Please see the definition of SPC in Appendix E. An SPC preference profile has a unique stable matching.

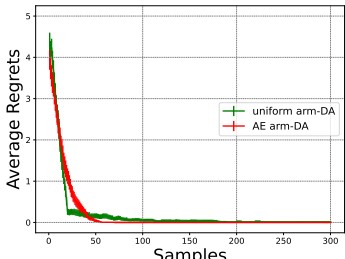 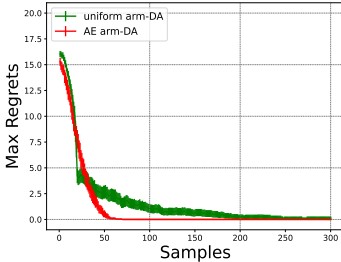

Figure 3: 95% confidence interval of agent-pessimal stable regrets for 200 randomized general preference profiles.

in secondary markets even when *all* preferences are known [Kojima et al., 2013]. We demonstrated key techniques in learning preferences that rely on the structure of stable solutions. In particular, exploiting the 'known' preferences of arms in the arm-proposing variant of DA and eliminating arms early on, provably reduces the sample complexity of finding stable matchings while experimentally having little impact on optimality (measured by regret). Findings of this paper can have substantial impact in designing new labor markets, school admissions, or healthcare where decisions must be made as preferences are revealed [Rastegari et al., 2014].

We conclude by discussing some limitations and open questions. First, extending this framework to settings with incomplete preferences, ties, or those that go beyond subgaussian utility assumptions are interesting directions for future research. We opted to avoid these nuances, for example ties, as such variations often introduce computational complexity with known preferences. In addition, given that the number of stable solutions could raise exponentially [Knuth, 1976], designing learning algorithms that could converge to stable solutions while satisfying some fairness notions (e.g. egalitarian or regret-minimizing) is an intriguing future direction.

## Acknowledgments and Disclosure of Funding

Hadi Hosseini acknowledges support from National Science Foundation (NSF) IIS grants #2144413 and #2107173. We thank Shraddha Pathak for her feedback. We also thank the anonymous reviewers for their comments and constructive criticisms.

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

# Appendix

## A    Related work (Extended)

**Stable matching markets.**    The two-sided matching market problem has been used to analyze many markets, such as the residency assignment problem [Gale and Shapley, 1962, Roth and Sotomayor, 1992]. The deferred-acceptance (DA) algorithm is an elegant procedure that guarantees a stable solution and can be computed in polynomial time [Gale and Shapley, 1962]. The DA algorithm is strategy proof for proposers Roth [1982], i.e., no single proposer can be matched to a better partner by misrepresenting the preference. However, proposers can form a coalition to misrepresent their preferences and some proposers are better off Huang [2006], Dubins and Freedman [1981]. Research on stable matching with uncertain preferences [Aziz et al., 2020, 2022] investigated to find a stable matching with the highest probability when both sides are unsure about their preferences.

**Bandit learning in matching markets.**    Liu et al. [2020] formalized the centralized and decentralized bandit learning problem in matching markets. In this problem, one side of participants (agents) have unknown preferences over the other side (arms), while arms have known preferences over agents. They introduced a centralized uniform sampling algorithm that achieves $O(\frac{K \log(T)}{\Delta^2})$ agent-optimal stable regret for each agent, considering the *time horizon $T$* and the minimum preference gap $\Delta$. For decentralized matching markets, Sankararaman et al. [2021] showed that there exists an instance such that the regret for agent $a_i$ is lower bounded as $\Omega(\max\{\frac{(i-1) \log(T)}{\Delta^2}, \frac{K \log(T)}{\Delta}\})$. Some research [Sankararaman et al., 2021, Basu et al., 2021, Maheshwari et al., 2022] focused on special preference profiles where a unique stable matching exists. Recently, Kong and Li [2023], Zhang et al. [2022] both unveiled decentralized algorithms to achieve a near-optimal regret bound $O(\frac{K log T}{\Delta^2})$, embodying the exploration-exploitation trade-off central to reinforcement learning and multi-armed bandit problems.

Other works focused on different variants of the bandit matching market problem. Das and Kamenica [2005], Zhang and Fang [2024] studied the bandit problem in matching markets, where both sides of participants have unknown preferences. Wang et al. [2022], Kong and Li [2024] generalized the one-to-one setting to many-to-one matching markets under the bandit framework. Kong and Li [2024] proposed an ODA algorithm that utilized a similar idea of arm-proposing DA variant compared to the AE arm-DA algorithm in this paper, however, the ODA algorithm achieved $O(\frac{NK}{\Delta^2} \log(T))$ regret bound in the one-to-one setting, which is $O(N)$ worse than the state-of-the-art algorithms in one-to-one matching (e.g. Kong and Li [2023], Zhang et al. [2022]). The performance of the ODA algorithm is hindered by unnecessary agent pulls. [Jagadeesan et al., 2021] studied stability of bandit problem in matching markets with monetary transfer. [Min et al., 2022] studied Markov matching markets by considering unknown transition functions.

## B    Omitted proofs from Section 3

**Theorem 1.** *Assume that the true preferences satisfy uniqueness consistency condition. For any estimated utility $\hat{\mu}$, if the agent-proposing DA algorithm produces a stable matching, then the arm-proposing DA algorithm produces a stable matching.*

*Proof.* We denote the matching generated based on the estimated utility $\hat{\mu}$ by the agent-proposing DA and arm-proposing DA as $\hat{m}_1$ and $\hat{m}_2$, respectively. By the definition of uniqueness consistency there is only one stable matching, denote it by $m^*$. Since $\hat{m}_1$ is assumed to be stable, $\hat{m}_1 = m^*$, we show that the arm-proposing DA algorithm also returns the same matching, i.e., $\hat{m}_2 = m^*$.

By using the Rural-Hospital theorem [Roth, 1986] on the estimated preferences $\hat{\mu}$, we have that the same subset of agents and arms are matched in both $\hat{m}_1$ and $\hat{m}_2$, so we can reduce the case to $N = K$ by only considering the subset of matched agents and arms.

Since the true preferences satisfy uniqueness consistency, then it must satisfy the $\alpha$-condition. Using the definition of $\alpha$-condition, we have an ordering of agents and arms such that

$$a_i \succ_{b_i} a_j, \forall j > i, \tag{2}$$

where $a_i = \hat{m}_1(b_i)$.

Suppose for contradiction that $\hat{m}_2 \neq m^*$. Then there must exist some $k > l$, such that $\hat{m}_2(b_l) = a_k$. However, since both $\hat{m}_1$ and $\hat{m}_2$ are stable with respect to $\hat{\mu}$ and $\hat{m}_2$ is arm optimal, the partner of arm $b_l$ in $\hat{m}_2$ is at least as good as its partner in $\hat{m}_1$. Thus,

$$a_k = \hat{m}_2(b_l) \succ_{b_l} \hat{m}_1(b_l) = a_l,$$

which is a contradiction to equation (2). Thus, we prove that $\hat{m}_2$ is also stable (with respect to true utility). $\qquad\square$

## C   Omitted proofs from Section 4

The following lemmas are useful to prove the stability bounds.

**Lemma 4** (Property of independent subgaussian, Lemma 5.4 in Lattimore and Szepesvári [2020]). *Suppose that $X$ is $d$-subgaussian and $X_1$ and $X_2$ are independent and $d_1$ and $d_2$ subgaussian, respectively, then we have the following property:*
*(1) $Var[X] \leq d^2$.*
*(2) $cX$ is $|c|d$-subgaussian for all $c \in \mathbb{R}$.*
*(3) $X_1 + X_2$ is $\sqrt{d_1^2 + d_2^2}$-subgaussian.*

By the property of independent subgaussian random variables, we have the following lemma that bounds the probability of ranking two arms wrongly.

**Lemma 5.** *Sample $h_1$ data $\{X_1, X_2, \ldots, X_{h_1}\}$ i.i.d. from 1-subgaussian with mean $\mu_1$, and $h_2$ data $\{Y_1, Y_2, \ldots, Y_{h_2}\}$ i.i.d. from 1-subgaussian with mean $\mu_2$, where $\mu_1 < \mu_2$. Two datasets are independent. Define $\hat{\mu}_1$ and $\hat{\mu}_2$ as the sample mean for two datasets. Then we have*

$$P(\hat{\mu}_2 < \hat{\mu}_1) \leq exp(-\frac{(\mu_2 - \mu_1)^2}{2(\frac{1}{h_1} + \frac{1}{h_2})}).$$

*Proof.* By Lemma 4, $\hat{\mu}_2 - \hat{\mu}_1$ is $\sqrt{\frac{1}{h_1} + \frac{1}{h_2}}$-subgaussian with mean $\mu_2 - \mu_1$. Thus by the definition of subgaussian

$$P(\hat{\mu}_2 < \hat{\mu}_1) = P((\hat{\mu}_2 - \hat{\mu}_1) - (\mu_2 - \mu_1) < -(\mu_2 - \mu_1)) \leq exp(-\frac{(\mu_2 - \mu_1)^2}{2(\frac{1}{h_1} + \frac{1}{h_2})}),$$

and the proof is complete. $\qquad\square$

**Lemma 1.** *Assume $\hat{\mu}$ is the sample average, and matching $\hat{m}$ is stable with respect to $\hat{\mu}$. Define a 'good' event for agent $a_i$ and arm $b_j$ as $\mathcal{F}_{i,j} = \{|\mu_{i,j} - \hat{\mu}_{i,j}| \leq \Delta/2\}$, and define the intersection of the good events over envy-set as $\mathcal{F}(ES(\hat{m})) = \cap_{(a_i, b_j) \in ES(\hat{m})} \mathcal{F}_{i,j}$. Then if the event $\mathcal{F}(ES(\hat{m}))$ occurs, matching $\hat{m}$ is guaranteed to be stable (with respect to $\mu$).*

*Proof.* We show that no agent-arm pair forms a blocking pair in $\hat{m}$. We prove it by contradiction.

Assume that there exists an agent $a_i$ and an arm $b_j$ in the local envy-set $ES_i(\hat{m})$ that blocks $\hat{m}$, which means $\mu_{i,\hat{m}(a_i)} < \mu_{i,j}$, and more concretely, $\mu_{i,j} - \mu_{i,\hat{m}(a_i)} \geq \Delta$. From stability of $\hat{m}$ with respect to the preference $\hat{\mu}$ we have that $\hat{\mu}_{i,\hat{m}(a_i)} > \hat{\mu}_{i,j}$, otherwise, $(a_i, b_j)$ blocks $\hat{m}$ according to $\hat{\mu}$. Thus, under the event $\mathcal{F}(ES(\hat{m}))$, it holds that

$$\hat{\mu}_{i,j} \geq \mu_{i,j} - \frac{\Delta}{2} \geq \mu_{i,\hat{m}(a_i)} + \frac{\Delta}{2} \geq \hat{\mu}_{i,\hat{m}(a_i)},$$

which is a contradiction. Therefore, $\hat{m}$ is stable with respect to $\mu$. $\qquad\square$

**Theorem 2.** *By uniform sampling (Algorithm 1), each agent samples each arm $T/K$ times, and $\hat{m}_1$ and $\hat{m}_2$ are matchings generated by agent-proposing DA and arm-proposing DA, respectively. Then,*

*(i) $P(\hat{m}_1 \text{ is unstable}) = O(|ES(\overline{m})| \exp(-\frac{\Delta^2 T}{8K}))$,*

*(ii) $P(\hat{m}_2 \text{ is unstable}) = O(|ES(\underline{m})| \exp(-\frac{\Delta^2 T}{8K}))$,*

*where $\underline{m}$ is the agent-pessimal stable matching and $\overline{m}$ is the agent-optimal stable matching.*

*Proof.* Since $\hat{m}_1$ and $\hat{m}_2$ are produced by DA based on $\hat{\mu}$, both matchings are stable with respect to $\hat{\mu}$. By Lemma 1, both matchings are guaranteed to be stable with respect to $\mu$ conditioned on $\mathcal{F}(ES(\hat{m}_1))$ (or $\mathcal{F}(ES(\hat{m}_2))$). Thus, it follows

$$
\begin{aligned}
P(\hat{m}_1 \text{ is unstable}) &\leq 1 - P(\mathcal{F}(ES(\hat{m}_1))) \\
&= \mathbb{E}[\cup_{(a_i, b_j) \in ES(\hat{m}_1)} P(|\mu_{i,j} - \hat{\mu}_{i,j}| \geq \Delta/2)] \\
&\leq \mathbb{E}[\sum_{(a_i, b_j) \in ES(\hat{m}_1)} P(|\mu_{i,j} - \hat{\mu}_{i,j}| \geq \Delta/2)] \\
&\leq 2\mathbb{E}[|ES(\hat{m}_1)|] \exp(-\frac{\Delta^2 T}{8K}),
\end{aligned}
$$

where the second line follows from the definition of the event $\mathcal{F}$, the third line is by union bound, and the last line follows from Lemma 4 that $\mu_{i,j} - \hat{\mu}_{i,j}$ is $\sqrt{\frac{K}{T}}$-subgaussian with 0 mean.

To complete the proof for $\hat{m}_1$, we demonstrate an upper bound of $\mathbb{E}[|ES(\hat{m}_1)|]$. Then we have that

$$
\begin{aligned}
|\mathbb{E}[|ES(\hat{m}_1)|] - |ES(\overline{m})|| &\leq NK \cdot P(\hat{m}_1 \neq \overline{m}) \\
&\leq NK \cdot P(\exists i, j, j' \text{ such that } \mu_{i,j} > \mu_{i,j'}, \hat{\mu}_{i,j} < \hat{\mu}_{i,j'}) \\
&\leq N^2 K^3 P(\mu_{i,j} > \mu_{i,j'}, \hat{\mu}_{i,j} < \hat{\mu}_{i,j'}) \\
&\leq N^2 K^3 exp(-\frac{\Delta^2 T}{4K}),
\end{aligned}
$$

where the second line comes from the fact that if $\hat{m}_1 \neq \overline{m}$, then there exists an agent $a_i$ and two arms $b_j$ and $b_{j'}$ that are learned incorrectly since a correct estimated profile produces $\overline{m}$ by agent-proposing DA. The last inequality follows from Lemma 5 since $(\mu_{i,j} - \mu_{i,j'}) \geq \Delta$ and number of samples is $T/K$.

Note that the difference is negligible when $T$ is sufficiently large compared with $N$ and $K$. The same computation applies to $\hat{m}_2$ and $\underline{m}$. Thus the second statement follows. $\qquad\square$

A useful technical lemma for bounding the number of samples agents need to find a stable matching is as follows.

**Lemma 6.** *For variables $K \in \mathbb{N}$ and $d > 0$, $\min\{t \in \mathbb{N} : log(Kt)/t \leq d\} = \Theta(\frac{1}{d} \log(\frac{K}{d}))$.*

*Proof.* Define $T_{\min} := \min\{t \in \mathbb{N} : log(Kt)/t \leq d\}$ and $T_{\max} := \max\{t \in \mathbb{N} : log(Kt)/t \geq d\}$. We observe that $T_{\min} \leq T_{\max} + 1$ and thus, we compute the upper bound of $T_{\min}$ through $T_{\max}$. By the definition of $T_{\max}$, we have

$$
T_{\max} \leq \frac{1}{d} \log(KT_{\max}) \leq \frac{1}{d} \log(\frac{K}{d} \log(KT_{\max})). \tag{3}
$$

Again by the definition of $T_{\max}$ and the fact that $x \geq log^2(x)$ for any $x > 0$, we have

$$
\frac{T_{\max}}{K} d^2 \leq \frac{\log^2(KT_{\max})}{KT_{\max}} \leq 1. \tag{4}
$$

Equation (3) and Equation (4) give

$$
T_{\max} \leq \frac{1}{d} \log(\frac{2K}{d} \log(\frac{K}{d})) = \frac{1}{d} \log(\frac{2K}{d}) + \frac{1}{d} \log(\log(\frac{K}{d})) \leq \frac{2}{d} \log(\frac{2K}{d}).
$$

On the other hand, we compute the lower bound of $T_{\min}$ by the definition of $T_{\min}$:

$$
T_{\min} \geq \frac{1}{d} log(KT_{\min}) \geq \frac{1}{d} \log(\frac{K}{d} \log(KT_{\min})).
$$

Since $T_{\min}$ is a positive integer, we have that

$$
T_{\min} \geq \frac{1}{d} \log(\frac{K}{d} \log(K)) \geq \frac{1}{d} \log(\frac{K}{d})
$$

when $K > 2$. Thus the proof is complete. $\qquad\square$

**Theorem 3.** *[Sample complexity for uniform sampling algorithm] With probability at least $1 - \alpha$, both the uniform agent-DA and the uniform arm-DA algorithms find a stable matching with the same sample complexity $\tilde{O}(\frac{NK}{\Delta^2} \log(\alpha^{-1}))$*[6].

*Proof.* We first show that Algorithm 1 finds the stable matching $\overline{m}$ with a high probability. Then we analyze the total number of samples. If agent $a_i$ samples arm $b_j$ for $t$ times, by Lemma 4 and the definition of subgaussian, with probability at least $1 - \frac{2}{(Kt)^\beta}$ such that $|\hat{\mu}_{i,j}(t) - \mu_{i,j}| \leq \sqrt{2\beta \log(Kt)/t}$, where $\hat{\mu}_{i,j}(t)$ is the sample average of agent $a_i$ over arm $b_j$ when $a_i$ samples $b_j$ for $t$ times. Taking a union bound for all $i, j, t$ gives that with probability at least $1 - \frac{2N}{K^{\beta-1}}\zeta(\beta)$,

$$|\hat{\mu}_{i,j}(t) - \mu_{i,j}| \leq \sqrt{2\beta \log(Kt)/t}, \forall t \in \mathbb{N}, \forall i \in [N], \forall j \in [K]. \tag{5}$$

Conditioned on this, we first show that Algorithm 1 terminates with true preference profiles. Assume that the mechanism stops with sample size $T$ for each agent-arm pair. For any $i$ and $j \neq k$, if $\hat{\mu}_{i,j} > \hat{\mu}_{i,k}$, we have a stopping condition

$$\hat{\mu}_{i,j} - \hat{\mu}_{i,k} \geq 2\sqrt{2\beta \log(KT)/T}, \tag{6}$$

since by Equation (1) and line 6 of Algorithm 1

$$LCB_{i,j} = \hat{\mu}_{i,j} - \sqrt{2\beta \log(KT)/T} \geq \hat{\mu}_{i,k} + \sqrt{2\beta \log(KT)/T} = UCB_{i,k}.$$

Then by Equation (5)

$$\mu_{i,j} \geq LCB_{i,j} \geq UCB_{i,k} \geq \mu_{i,k}.$$

Hence, uniform agent-DA algorithm produces the stable matching $\overline{m}$.

Then we compute the sample complexity. By Equation (5)

$$\hat{\mu}_{i,j} - \hat{\mu}_{i,k} \geq -2\sqrt{2\beta \log(KT)/T} + \mu_{i,j} - \mu_{i,k}. \tag{7}$$

Therefore if we set

$$T = \min\{t \in \mathbb{N} : \sqrt{2\beta \log(Kt)/t} \leq \Delta/4\},$$

we have that

$$\begin{aligned}
\hat{\mu}_{i,j} - \hat{\mu}_{i,k} &\geq -2\sqrt{2\beta \log(KT)/T} + \mu_{i,j} - \mu_{i,k} \\
&\geq -2\sqrt{2\beta \log(KT)/T} + \Delta \\
&\geq 2\sqrt{2\beta \log(KT)/T}.
\end{aligned}$$

and thus the stopping condition 6 is satisfied. By Lemma 6 we have $T = \min\{t \in \mathbb{N} : \sqrt{2\beta \log(Kt)/t} \leq \Delta/4\} = O(\frac{\beta}{\Delta^2} \log(\frac{\beta K}{\Delta^2}))$. Note that $T$ is the sample complexity for each agent-arm pair. Thus, the total number of samples are bounded by $NKT = O(\frac{\beta NK}{\Delta^2} \log(\frac{\beta K}{\Delta^2}))$.

By setting a probability budget $\alpha = \frac{4N}{K^{\beta-1}}$, we have that with probability at least $1 - \alpha$, the uniform sampling algorithm terminates with sample complexity $O(\frac{\beta NK}{\Delta^2} \log(\frac{\beta K}{\Delta^2}))$, where $\beta = 1 + \frac{\log(4N\alpha^{-1})}{\log(K)}$. Therefore, the sample complexity for uniform agent-DA algorithm is $\tilde{O}(\frac{NK}{\Delta^2} \log(\alpha^{-1}))$.

For uniform arm-DA, the only difference is that it produces a matching by arm-proposing DA algorithm. Therefore, uniform arm-DA finds the stable matching $\underline{m}$ with the same sample complexity. $\square$

# D   Omitted proofs from Section 5

**Theorem 4.** *By Algorithm 3, assume that agent $a_i$ samples arm $b_j$ for $T_{i,j}$ and $\hat{m}$ is returned by the algorithm. We define $T_{min} = \min_{(a_i,b_j) \in ES(\hat{m})} T_{i,j}$ as the minimum sample size for agent-arm pairs. Then, we have*

$$P(\hat{m} \text{ is unstable}) \leq O(|ES(\underline{m})| \exp(-\frac{\Delta^2 T_{min}}{8})).$$

---

[6]$\tilde{O}$ denotes the upper bound that omits terms logarithmic in the input.

*Proof.* When the sampling phase of Algorithm 3 ends, there is no unmatched agent and the matching $\hat{m}$ is the same matching proposed by arm-proposing DA according to estimated utility $\hat{\mu}$. Thus, $\hat{m}$ is stable with respect to $\hat{\mu}$. By Lemma 1, we know that $\hat{m}$ is stable under the event $\mathcal{F}(ES(\hat{m}))$. Then

$$
\begin{aligned}
P(\hat{m} \text{ is unstable}) &\leq 1 - P(\mathcal{F}(ES(\hat{m}))) \\
&= \mathbb{E}[\cup_{(a_i, b_j) \in ES(\hat{m})} P(|\mu_{i,j} - \hat{\mu}_{i,j}| \geq \Delta/2)] \\
&\leq \mathbb{E}[\sum_{(a_i, b_j) \in ES(\hat{m})} P(|\mu_{i,j} - \hat{\mu}_{i,j}| \geq \Delta/2)] \\
&\leq 2\mathbb{E}[\sum_{(a_i, b_j) \in ES(\hat{m})} \exp(-\frac{\Delta^2 T_{i,j}}{8})] \\
&\leq 2\mathbb{E}[|ES(\hat{m})| \exp(-\frac{\Delta^2 T_{min}}{8})].
\end{aligned}
$$

The third line comes from union bound, and the fourth line comes from Lemma 4 and that $\mu_{i,j} - \hat{\mu}_{i,j}$ is $\sqrt{\frac{1}{T_{i,j}}}$-subgaussian. Observe that correct estimated profile on $ES(\hat{m})$ outputs $\underline{m}$ by Algorithm 3 since the algorithm follows arm-DA algorithm. Then we have the computation

$$
\begin{aligned}
P(\hat{m} \neq \underline{m}) &\leq P(\exists (a_i, b_j) \in ES(\hat{m}), (a_i, b_{j'}) \in ES(\hat{m}), \text{ and } \mu_{i,j} > \mu_{i,j'}, \hat{\mu}_{i,j} < \hat{\mu}_{i,j'}) \\
&\leq \sum_{(a_i, b_j), (a_i, b_{j'}) \in ES(\hat{m})} P(\mu_{i,j} > \mu_{i,j'}, \hat{\mu}_{i,j} < \hat{\mu}_{i,j'}) \\
&\leq \sum_{(a_i, b_j), (a_i, b_{j'}) \in ES(\hat{m})} exp(-\frac{\Delta^2}{2(\frac{1}{T_{i,j}} + \frac{1}{T_{i,j'}})}) \\
&\leq NK^2 exp(-\frac{\Delta^2 T_{min}}{4}),
\end{aligned}
$$

where the third line follows from Lemma 5. Therefore, we have that $\hat{m}$ converges to $\underline{m}$ and so the probability of not being $\underline{m}$ is negligible when $T_{min}$ is sufficiently large. Thus, we complete the proof. $\square$

**Theorem 5.** *[Sample complexity for AE arm-DA algorithm] With probability at least $1 - \alpha$, Algorithm 3 terminates and returns a stable matching, $\underline{m}$, with sample complexity of*

$$
\tilde{O}(\frac{ES(\underline{m})}{\Delta^2} \log(\alpha^{-1})).
$$

*Proof.* The proof has similar structure to the proof of Theorem 3; however, the proof is different because Algorithm 3 only sample the pairs in the envy set $ES(\underline{m})$ while Algorithm 1 samples all arms uniformly.

We first show that with a high probability, the algorithm terminates with $\underline{m}$. If agent $a_i$ samples arm $b_j$ for $t$ times, by Lemma 4 and the definition of subgaussian, we have that $|\hat{\mu}_{i,j}(t) - \mu_{i,j}| \leq \sqrt{2\beta \log(Kt)/t}$ with probability at least $1 - \frac{2}{(Kt)^\beta}$, here $\mu_{i,j}(t)$ is the sample average of agent $a_i$ over arm $b_j$ when $a_i$ samples $b_j$ for $t$ times. Taking a union bound over all $t$ and all pairs $a_i, b_j$ in the envy set $ES(\underline{m})$, we have that

$$
|\hat{\mu}_{i,j}(t) - \mu_{i,j}| \leq \sqrt{2\beta \log(Kt)/t}, \forall t \geq 1, \forall (a_i, b_j) \in ES(\underline{m}) \tag{8}
$$

with probability at least $1 - \frac{2|ES(\underline{m})|}{K^\beta} \zeta(\beta)$. Conditioned on Equation (8), we have that Algorithm 3 terminates with correct estimated profile on pairs of the envy-set $ES(\underline{m})$ as the same logic in the first part of the proof in Theorem 3. Thus, since Algorithm 3 follows arm-proposing DA algorithm, the algorithm outputs $\underline{m}$.

Next, we claim that for every agent-arm pair in the envy-set $ES(\underline{m})$, the sample complexity is $T = \min\{t \in \mathbb{N} : \sqrt{2\beta \log(Kt)/t} \leq \Delta/4\} = O(\frac{\beta}{\Delta^2} \log(\frac{\beta K}{\Delta^2}))$. We prove this for each agent-arm pair by induction on the number of iterations of while loop of Algorithm 3. We will show that in each invocation of Algorithm 2, the number of samples for any agent-arm pair in the envy-set $ES(\underline{m})$ is at most $T$. Thus, since the while loop in Algorithm 3 checks whether the sample number is bounded by $T$, we get the desired bound.

**Base case.** Let arms $b_j$ and $b_{j'}$ are the first to propose to agent $a_i$. Let $T_{i,j}$ and $T_{i,j'}$ be the number of samples for agent $a_i$ over arm $b_j$ and $b_{j'}$ by Algorithm 2. Without loss of generality, we assume that $\mu_{i,j} > \mu_{i,j'}$. Since both arms have not been sampled before, by Algorithm 2 the stopping condition Equation (6) is satisfied when $T_{i,j} = T_{i,j'} = T$ since

$$\hat{\mu}_{i,j} - \hat{\mu}_{i,j'} \geq -2\sqrt{2\beta \log(KT)/T} + \mu_{i,j} - \mu_{i,j'}$$
$$\geq -2\sqrt{2\beta \log(KT)/T} + \Delta$$
$$\geq 2\sqrt{2\beta \log(KT)/T},$$

where the first line comes from Equation (8), and the third line comes from the definition of $T$.

**Inductive steps.** By induction hypothesis, in Algorithm 3, agent $a_i$ has already sampled $b_j$ for $t_{i,j} \leq T$ times. Let $b_j$ be the winner in the last round and $b_{j'}$ proposes to $a_i$ in this round. In Line 4 of Algorithm 2, agent $a_i$ samples $t_{i,j}$ times of arm $b_{j'}$ before sampling $b_j$. Then if $a_i$ samples $b_j$ for $T - t_{i,j}$ more times and $a_i$ samples $b_{j'}$ $T$ times, by the same computation as the base case, we have that the stopping condition Equation (6) is satisfied when $T_{i,j} = T_{i,j'} = T$. Thus, we show that the number of samples for any agent-arm pair is bounded by $T$.

Since Algorithm 3 only samples the agent-arm pairs in the envy set $ES(\underline{m})$, we get that the total sample complexity is $|ES(\underline{m})|T = O(\frac{\beta(|ES(\underline{m})|)}{\Delta^2} \log(\frac{\beta K}{\Delta^2}))$.

By setting the probability budget $\alpha = \frac{4|ES(\underline{m})|}{K^\beta}$, we have that with probability at least $1 - \alpha$, the AE arm-DA algorithm has sample complexity complexity of the order $\frac{\beta|ES(\underline{m})|}{\Delta^2} \log(\frac{\beta K}{\Delta^2})$, where $\beta = \frac{\log(4|ES(\underline{m})|\alpha^{-1})}{\log(K)}$. Therefore, it is of the order $\tilde{O}(\frac{|ES(\underline{m})|}{\Delta^2} \log(\alpha^{-1}))$. $\qquad\square$

**Lemma 3.** *Considering any true preference $\mu$, we have the following bounds for envy-set:*

*(i) Size of the envy-set for $\overline{m}$: $(max\{N, K\} - N)N \leq |ES(\overline{m})| \leq NK$.*

*(ii) Size of the envy-set for $\underline{m}$: $(max\{N, K\} - N)N \leq |ES(\underline{m})| \leq NK - N + 1$.*

*Proof.* First we consider the best case and $N \geq K$. Suppose that for any $i \in [K]$, agent $a_i$ prefers arm $b_i$ the most; also for any $j \in [K]$, arm $b_j$ prefers agent $a_j$ the most. Under this preference profile, we immediately have $\overline{m} = \underline{m} = \{(a_1, b_1), (a_2, b_2), \ldots, (a_K, b_K)\}$ and, since all arms match their first choice, we have $|ES(\overline{m})| = |ES(\underline{m})| = 0$. if $K > N$, we construct the preference profile similarly for the first $N$ agents and $N$ arms, then the remaining $K - N$ arms are not matched, and so $|ES(\overline{m})| = |ES(\underline{m})| = (K - N)N$.

Now consider the worst case for $\overline{m}$. We keep agents' preferences the same as stated above; however, agent $a_j$ is the worst agent according to $b_j$ for each $j \in [min\{N, K\}]$. Then $\overline{m} = \{(a_1, b_1), (a_2, b_2), \ldots, (a_{min\{N,K\}}, b_{min\{N,K\}})\}$. Thus, for each $j \in [K]$, arm $b_j$ is in the envy-set $ES_i(\overline{m})$ of agent $a_i$ for all $i \in [N]$. Then, envy-set $ES(\overline{m})$ contains all agent-arm pairs and so in the worst case $|ES(\overline{m})| = NK$.

Lastly, we show the worst case for $\underline{m}$. We claim that for all matched arms, at most one arm can get matched to the worst choice. In their seminal work, Gale and Shapley [1962] observed that when one arm gets matched to the worst agent, it proposes to all other agents and get rejected by them. This observation implies all other agents are tentatively matched and thus no agent is available. Therefore, we have that the worst case is $|ES(\underline{m})| = NK - N + 1$. $\qquad\square$

# E  Omitted details from Section 6

We define the sequence preference condition (SPC) that we use for our experiments. A preference profile satisfies SPC if and only if there is an order of agents and arms such that $\forall i \in [N], \forall j > i, \mu_{i,i} > \mu_{i,j},$ *and* $\forall i \in [K], \forall j > i, a_i >_{b_i} a_j$. If each participant of one side (agents or arms) has the same preference (also known as a masterlist ) over the other, the preference profile satisfies SPC. Clearly, a preference profile that is SPC satisfies $\alpha$-condition.

Figure 4 shows the stability and regrets for all four algorithms under the constraint of the agent masterlist preference profiles. When agents share identical utilities for each arm, the average rewards

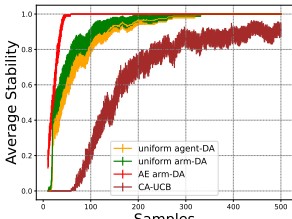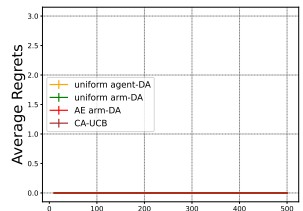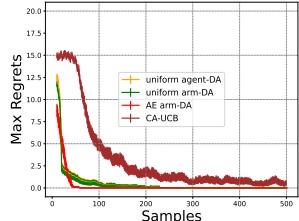

Figure 4: 95% confidence interval of stability and regrets for 200 randomized agent masterlist preference profiles.

remain the same regardless of the matching. Consequently, the average regrets (showed in the middle figure) are always zero.

We also explain why the uniform arm-DA algorithm surpasses the AE arm-DA algorithm in terms of $\overline{R}$, but underperforms compared to the AE-arm DA algorithm in terms of $\underline{R}$ under general preference profiles. Note that the uniform arm-DA algorithm samples arms uniformly, and thus it may end up matching agents to arms that are between the agent-optimal stable match and the agent-pessimal stable match. Thus, when compared with the agent-pessimal regret, it may seem better. However, in comparison, agents are incrementally matched to increasingly preferable arms throughout the procedure of Algorithm 3, and therefore agents are matched to arms that are no better than agent-pessimal stable match as shown in Figure 1 and Figure 3.

