# OpenReview forum: "Putting Gale & Shapley to Work: Guaranteeing Stability Through Learning"
_NeurIPS.cc/2024/Conference — NeurIPS 2024 poster_

### Official Review · Reviewer_ct3v · 2024-07-11

**Soundness:** 3
**Presentation:** 3
**Contribution:** 2
**Rating:** 5
**Confidence:** 5

**Summary:**

This paper investigates the bandit learning problem in matching markets. It introduces a critical perspective that the objective of regret minimization does not align with achieving market stability. The study explores the sample complexity required to find a stable matching. To address this issue, the authors propose two algorithms: the Uniform Sampling DA algorithm and the Arm Elimination DA algorithm, and provide guarantees based on the corresponding envy-set size.

**Strengths:**

1. This paper raises a significant issue in the literature of bandit learning in matching markets, highlighting that the objective of regret minimization may not align well with the goal of finding a stable matching.

2. The investigation of sample complexity in this paper delves into the detailed structure, specifically examining the envy-set size with respect to a particular stable matching, thereby deepening the understanding of the learning task's difficulty.

**Weaknesses:**

1.Although some existing works consider stable regret, their theoretical guarantees also imply guarantees for the sample complexity of reaching (player-optimal/pessimal) stable matchings, as demonstrated by Liu et al. [2020], Kong and Li [2023], and Zhang et al. [2022]. However, this paper does not compare its results with the induced sample complexity of these existing works. Consequently, it is difficult to determine how this work improves upon existing studies in terms of sample complexity and the objective of achieving stable matchings, specifically player-optimal and player-pessimal stable matchings.

2.The technical novelty is limited. The uniform-sampling DA algorithm is standard in the literature, and the arm elimination DA algorithm appears similar to the ODA algorithm in Kong and Li [2024], although they consider a more general many-to-one setting. Please verify this.

**Questions:**

Please see the last part.

**Limitations:**

The authors adequately addressed the limitations.

---

> ### Author Rebuttal · Authors · 2024-08-06
>
> We thank the reviewer for the comments. Please see our responses and clarifications below.
>
> > Although some existing works consider stable regret, their theoretical guarantees also imply guarantees for the sample complexity of reaching (player-optimal/pessimal) stable matchings, as demonstrated by Liu et al. [2020], Kong and Li [2023], and Zhang et al. [2022]. However, this paper does not compare its results with the induced sample complexity of these existing works.
>
>
> Response: We do compare our approach with those of Liu et al. [2021] and  Kong and Li [2023]. These are shown in plots as CA-UCB (in Line 326) and uniform agent-DA, respectively. We will mention it explicitly.  Please note that  Kong and Li [2023] provided an ETGS algorithm, which is essentially the same as the uniform agent-DA algorithm in this paper with minor differences. Both algorithms take UCB/LCB bounds to indicate when to stop sampling. ETGS algorithm has an extra phase of index estimation that only takes $N^2$ samples so the sample complexity in this phase can be ignored, and therefore, the sample complexity of ETGS algorithm would have the same order as uniform agent-DA in our paper.
>
> >Consequently, it is difficult to determine how this work improves upon existing studies in terms of sample complexity and the objective of achieving stable matchings, specifically player-optimal and player-pessimal stable matchings.
>
> Response: One of the main advantages of this paper is to emphasize the role of arm-proposing DA, while many previous papers including Liu et al. [2020], Kong and Li [2023], take the agent-proposing DA. Action Elimination Arm-proposing DA algorithm takes the advantage of fewer sample complexity for a stable matching, as compared to solutions that take uniform agent-DA.
>
> > The technical novelty is limited. The uniform-sampling DA algorithm is standard in the literature, and the arm elimination DA algorithm appears similar to the ODA algorithm in Kong and Li [2024], although they consider a more general many-to-one setting. Please verify this.
>
> Response: The arm elimination DA algorithm is not the same as ODA. After careful comparison of the ODA algorithm, uniform sampling algorithms, and our AE arm-DA algorithm, we explain the significant differences below.
>
> (i) Firstly we would like to compare regret performance. As the reviewer also mentioned, the ODA algorithm was designed for many-to-one matching markets. Since one-to-one setting is a special case of many-to-one matching markets, we could recover the player-pessimal stable regret for the ODA algorithm as $\frac{NK}{\Delta^2}logT$ in the one-to-one setting. On the other hand, ETGS algorithm in Kong and Li [2023] (or uniform agent-DA algorithm in this paper) has player-optimal stable regret $\frac{K}{\Delta^2}logT$. Since the uniform arm-DA algorithm aims to reach the player-pessimal stable matching, we can show that its player-pessimal stable regret is also $\frac{K}{\Delta^2}logT$ without much difficulty. Therefore, the ODA algorithm has $N$ times the regret bound of the uniform arm-DA algorithm.
>
> Our simulated experiments show that the AE arm-DA algorithm converges with fewer samples in terms of player-pessimal stable regrets compared with uniform arm-DA algorithm, as shown in Figure 3. By fixing sample size $T$, we also compare regrets. Therefore, our AE arm-DA algorithm has lower regrets than that of the uniform arm-DA algorithm. Therefore, we get the following ordering of player-pessimal stable regret in one to one setting:
> $\underline{R}(ODA) > \underline{R}(uni \ arm-DA) > \underline{R}(AE \ arm-DA)$.
>
> (ii) More importantly, we would like to emphasize that the technical novelty of this paper is the notion of envy-set. While many previous papers including Liu et al. [2020, 2021], Kong and Li [2023, 2024] constructed their theory based on the number of agents $N$ and the number of arms $K$, we discover that the difficulty of the learning problem could depend on a new notion, i.e. envy-set. The ODA algorithm along with many others failed to capture this important observation.

---

> ### Comment · Reviewer_ct3v · 2024-08-10
>
> Thanks for your detailed response. I am happy to increase my score. I also want to further ask how to derive an $O(K\log T/\Delta^2) $ regret for the AE-arm-DA? As its current result depends on $|ES(\underline{m})|$ which is of order $O(NK)$.

---

> ### Author Response · Authors · 2024-08-10
>
> Thank you for taking the time to read our response and seek clarification. We are glad to clarify your question.
>
> Please note that the definition of player-pessimal stable regret is the regret for only ONE agent, while the definition of sample complexity in our paper is how many samples ALL agents need collectively to reach a stable matching. Therefore, uniform arm-DA algorithm has player pessimal stable regret depending on $K$ and sample complexity depending on $NK$; AE arm-DA algorithm has sample complexity depending on $|ES(\underline{m})|$.
>
> We didn't do regret analysis for AE arm-DA in this paper. Note that the ordering we wrote for player-pessimal stable regret is based on the empirical observations.

---

> > ### Comment · Reviewer_ct3v · 2024-08-10
> >
> > Thanks for your response. I wonder whether there is some misalignment between the stable regret and sample complexity. Consider the global preference example where all agents' preference rankings are $b_1>b_2>...>b_K$, and all arms' preference rankings are $a_1> a_2> ... >a_N$. In this case, the agent $a_1$ will sample all arms until it identifies $b_1$ is optimal. Then the agent $a_2$ will sample arms except for $b_1$ until it identifies $b_2$ to be optimal. The agent $a_N$ needs to wait for all of the other agents to sample arms. This part of the samples contributes not only to the regret of the agents with higher priority but also $a_N$'s own regret. However it does not contribute to the sample complexity repeatedly. So the sample complexity/N is not equal to a single agent's stable regret. Is this right?

---

> > > ### Author Response · Authors · 2024-08-10
> > >
> > > Right, it is not true that sample complexity/N is regret. In fact, as we saw, sample complexity is a function of probability budget $\alpha$, and regret is a function of $T$.
> > > Observe that we define regrets based on the difference between the sampled arm with the player-optimal (or pessimal) stable arm. When an agent does not sample an arm, it does not incur extra regret. In your example, by our AE arm-DA algorithm, when agent $a_1$ samples all arms, other agents are not sampling so they should have 0 regret.

---

> > > > ### Comment · Reviewer_ct3v · 2024-08-11
> > > >
> > > > I think it is not the traditional stable regret definition as it does not consider the collision scenario in the multi-agent problem. When a player samples nothing, it receives 0 rewards. So the regret is the mean of the (optimal or pessimal) stable arm, instead of 0. This is the definition in all of the existing works [Liu et al. 20; Kong and Li. 2023, 2024].

---

> ### Author Response · Authors · 2024-08-11
>
> Yes, We agree with the reviewer that our setting is a little different from the traditional setting and we do not consider regrets in our theory. Instead, we consider sample complexity, which focuses on minimizing the amount of samples to reach a stable matching.

---

> > ### Comment · Reviewer_ct3v · 2024-08-11
> >
> > Overall, I appreciate the contribution of studying the pure exploration problem in matching markets and the use of the envy-set notion to measure its complexity. However, since both the uniform sampling and arm elimination algorithms are similar to existing approaches, I am adjusting my score to 5. I encourage the authors to discuss the relationship between their sample complexity and the standard regret, as well as their algorithmic difference between arm elimination and ODA in the next version.

---

### Official Review · Reviewer_ormW · 2024-07-15

**Soundness:** 3
**Presentation:** 3
**Contribution:** 3
**Rating:** 7
**Confidence:** 3

**Summary:**

The paper studies stability in matching problems with learned preferences on one side. The two sides are agents and arms. The agents learn their preferences over the arms through sampling while the arms know their preferences over agents. Then a DA algorithm is run with either arms or agents proposing. The paper gives bounds on the probability that the matching found is unstable, as well as sample complexity results, with uniform sampling and for a more refined arm-elimination algorithm. The paper is complemented by numerical simulations.

**Strengths:**

The problem studied, the stability of the matching found by learning, is interesting and novel. Indeed, much of the literature rather focused on the regret achieved but I have not seen many works on looking at the probability of blocking pairs.

The bounds proved are relevant and well behaved, and relatively intuitive.

The comparison of uniform sampling and arm elimination (which basically removes a factor K) is nice.

The paper is easy to follow.

**Weaknesses:**

1. The bounds proved contain a term in |ES(m)|. This term is natural but may be quite large. It would be good to give an order of magnitude for some reasonable preferences.

2. The theoretical tools to prove the main results are relatively standard from bandits. This is not a weakness in itself, I am mentioning it to justify that I do not see the contribution as a particularly intricate theoretical analysis.

3. Is it possible to provide lower bounds to know whether the analysis is tight? And to do better than arm elimination?

4. The motivation for the problem studied could be further developed to justify the setting introduced (sampling, then DA). For instance, a reasonable alternative would be to have a more dynamic setting where different agents get matched at different times, e.g., when they are confident that they are done exploring (and then the pair leaves the system). Of course, this would be a different problem and I am not suggesting the authors should have studied this one instead, but it would be nice to strengthen the motivation for the given problem studied.

**Questions:**

See above, in particular 1. and 3.

---

> ### Author Rebuttal · Authors · 2024-08-06
>
> Thank you for your constructive comments. Please see the following responses to your questions.
>
> > The bounds proved contain a term in |ES(m)|. This term is natural but may be quite large. It would be good to give an order of magnitude for some reasonable preferences.
>
> Response: Please note that in the proof of Lemma 3, we constructed preference profiles that have the smallest envy-set size $|ES(\underline{m})| = (K-N)N$ (which would be reduced to $0$ if $N = K$) and the largest envy-set size $|ES(\underline{m})| = NK - N + 1$.
>
> One way to illustrate the magnitude of $|ES(\underline{m})|$ is through the lens of arm-proposing DA algorithm. Usually in a highly competitive environment for arms, e.g. there are much more arms than agents so that a lot of arms are not matched, the magnitude of $|ES(\underline{m})|$ is large since we need a large number of proposals and rejections in the process of arm-proposing DA. A less competitive environment, e.g. arms put different agents as their top choices, has much smaller magnitude of $|ES(\underline{m})|$.
>
>
> > The theoretical tools to prove the main results are relatively standard from bandits. This is not a weakness in itself, I am mentioning it to justify that I do not see the contribution as a particularly intricate theoretical analysis.
>
> Response: We agree with the reviewer that some of the theoretical techniques are borrowed from bandit literature. Please note that the notion of envy-set itself is novel, and we are able to construct our sample complexity based on it. Focusing on $N$ and $K$, as done in previous work, to achieve such bounds does not provide any insight about the preference structure or the structure of the stable matchings.
>
> > Is it possible to provide lower bounds to know whether the analysis is tight? And to do better than arm elimination?
>
> Response: This is an interesting question that we are also considering. We do not know of any lower bounds. We have an initial idea on whether it is possible to do better than arm elimination arm-proposing DA. Remember that uniform sampling algorithm has sample complexity $\frac{NK}{\Delta^2}log(\alpha^{-1})$ (see Theorem 3). Intuitively, $NK$ comes from the observation that agents uniformly pull all arms so that the number of agent-arm pairs that interact are $NK$. In Arm Elimination arm-proposing algorithm, there is a high probability that the number of agent-arm pairs that interact are $|ES(\underline{m})|$.
> Then we might ask this question: can we design a new algorithm that with high probability, the number of agent-arm pairs that interact are smaller than $|ES(\underline{m})|$?
>
>
> > The motivation for the problem studied could be further developed to justify the setting introduced (sampling, then DA). For instance, a reasonable alternative would be to have a more dynamic setting where different agents get matched at different times, e.g., when they are confident that they are done exploring (and then the pair leaves the system). Of course, this would be a different problem and I am not suggesting the authors should have studied this one instead, but it would be nice to strengthen the motivation for the given problem studied.
>
>
> Response: Thank you for your suggestion. Yes, one advantage of our solution as compared to the agent-proposing DA, such as Kong et al. 2023, is that agents do not need to explore arms concurrently.  I have a question about your dynamic model. Are you suggesting that agents and arms can come and leave the system? If they leave the system, I'm concerned that the matching might not be stable.

---

> > ### Comment · Reviewer_ormW · 2024-08-08
> >
> > Thank you for the responses, I am happy to keep my score as is.

---

### Official Review · Reviewer_wAZr · 2024-07-16

**Soundness:** 2
**Presentation:** 3
**Contribution:** 1
**Rating:** 4
**Confidence:** 4

**Summary:**

The paper studies the sample complexity of finding a stable matching under the probably approximately (PAC) framework.

In the model, $N$ agents are to be matched with $K \ge N$ arms. Each arm has an (a priori unknown) utility to each agent, inducing the agents’ preferences; Arms also has strict preferences over the agents. Agents learn about their rewards by pulling the arms and observing stochastic rewards with means equal to the utilities. The objective is to find a matching that is stable with the desired probability with least possible samples (pullings of arms).

The paper proposes an arm-proposing deferred acceptance (DA) algorithm with action elimination (AE). The algorithm implements the arm-proposing DA, with the modification where, in face of uncertain preferences over a new proposal and the current match, the agent uses AE to reach a decision of rejection. The sample complexity of the algorithm is analyzed and simulations are used to evaluate the algorithm empirically.

**Strengths:**

The paper combines the online learning from stochastic observations with the classic literature on stability of matchings, a novel and important direction. While intuitive, the algorithmic combination of arm-proposing DA with AE is novel to my understanding.

The paper is overall clearly written, with consistent notations and clear definitions. I find it easy to follow. The claims seem reasonable, although I did not go over all the proofs.

**Weaknesses:**

While the results may be new and interesting, I do not find them sufficiently significant. From the theoretical perspective, the algorithm proposed and its analysis both seem fairly standard. The algorithmic improvement also seem marginal, as Theorem 5 only seem to provide a weak improvement on existing results, and nor do the simulations (which are run in a very restrictive setting with a specific form of preferences) seem conclusive on the superiority of the main algorithm. [Edit: From the discussion, I understand that there is some valid rationales behind the simulations (the choice of utility distribution in particular). I think it is worth more discussion and justification.]

On the empirical front, the paper would benefit from additional motivating examples on why the results can be impactful in real world applications. As will be mentioned in one of my questions, I am not sure how the proposed algorithm can be applied to settings such as school admission, ride share, etc. - in particular, the assumption that rewards are iid and that an arm can be pulled (sampled) repeatedly in an arm-proposing fashion seems hard to interpret in these settings. Without sufficient practical motivation, the results seem only a nice yet marginal theoretical exercise.

**Questions:**

- Can the authors elaborate on the assumption that $N \le K$? Note that, since the main algorithm is an arm-proposing DA variant, the two sides are not symmetric and such an assumption requires justification. When $N > K$ we know that the entire set of stable matchings can exhibit strikingly different behaviors than when $N\le K$ [1].
- Is there any reason why the experiment is only carried out on the specific utility model with utilities 1 through 20? What if they come from other distributions, e.g., exponentially or polynomially growing? The additional SPC setting is also quite restrictive in my understanding.
- Have the authors considered relaxing the notion of stability in such stochastic settings (e.g., [2,3])? Ultimately, strict stability is a rather restricted solution concept and often beyond hope, and an approximate notion may be both more realistic and also more tractable.
- How should one reason about the minimum preference gap? In general, I would expect this quantity to scale down as the market size increases; realistically speaking, at most $1/n$.
- Further, indifference can be common in real world applications, and I would hope that the model and the algorithm should allow for such scenarios - pressing a student to form a precise preference over schools that are otherwise entirely comparable seems unnecessary. Maybe approximate stability could be one way to resolve this? E.g., maybe there is a better measure of “average stability” that accounts for utility gap rather than just the probability of stability (Line 324, in my understanding).

[1] Itai Ashlagi, Yash Kanoria, and Jacob D. Leshno. 2017. Unbalanced Random Matching Markets: The Stark Effect of Competition." Journal of Political Economy.
[2] Yannai A. Gonczarowski, Noam Nisan, Rafail Ostrovsky, and Will Rosenbaum. 2015. A Stable Marriage Requires Communication. In Proceedings of the 2015 Annual ACM-SIAM Symposium on Discrete Algorithms (SODA).
[3] Itai Ashlagi, Mark Braverman, and Geng Zhao. 2023. Welfare Distribution in Two-sided Random Matching Markets. In Proceedings of the 24th ACM Conference on Economics and Computation (EC '23).

Minor comments:
- Line 135: typo. “… leads to unique …” => “… leads to a unique …”
- Line 359-360: “such variations often introduce computational complexity with known preferences” - I might be missing something obvious, but in the hope to understand this claim better, I hope the authors could clarify what “computational complexity with known preferences” they are referring to here. In my understanding, with known preferences, the vanilla DA should work fine with ties and is efficient.
- When the problem is formulated as agents pulling arms, I find it hard to reasonable arms proposing. To me, this means that the side that bears the burden of exploration in fact does not control their very own action? Is there a better setup or application that motivates the study?

**Limitations:**

Limitations are discussed in Section 7: Conclusion and Future Work. The paper is mostly of a theoretical nature.

---

> ### Author Rebuttal · Authors · 2024-08-06
>
> We thank the reviewer for the comments and suggestions. We first clarify some of the comments.
> > The algorithmic improvement seem marginal, as Theorem 5…
>
> Please note that sample complexity of the AE arm-DA ($\frac{|ES(\underline{m})|}{\Delta^2}log(\alpha^{-1})$) has an improvement on that of uniform agent-DA ($\frac{NK}{\Delta^2}log(\alpha^{-1})$). The improvement ratio is $\frac{NK}{|ES(\underline{m})|}$. In the best case, the envyset $ES(\underline{m})$ has size $0$ when $N=K$ (see Remark 3) giving an infinitely better solution. This is the reason why we stress the role of arm-proposing DA in terms of stability. Moreover, the AE algorithm uses the envyset effectively (which has never been done before), unlike the uniform algorithms, as explained, leading to a significant improvement in the sample complexity.
> > Line 359-360: … the vanilla DA should work fine with ties and is efficient.
>
> When preferences are ordinal and contain ties, stable solutions may not exist in their strong sense (see, e.g. Irving et al. [1994], Manlove [2002]). We need to consider new stability notions (strict, strong, and weak stability). Moreover, in the presence of ties and incomplete lists, not all agents are matched in every weakly stable matching. In fact, finding a stable solution that is efficient (i.e., matches maximum number of agents) is NP-hard as shown in “Hard variants of stable matchings”, Manlove et al, TCS 2002.
> >When the problem is formulated as agents pulling arms, I find it hard to reasonable arms proposing.
>
> Please note the difference between “agents pulling arms” and “arms proposing to agents”. The former is our model setup, while the latter is a step in our algorithm. The goal of the problem is to minimize the exploration agents spend in finding a stable matching. Either side of the matching market can propose depending on the algorithm we use.
> An example in labor markets: Companies are agents and job seekers are arms. When a company organizes an interview with a potential employee, it incurs cost. A company usually does not have prior information about a job seeker in advance. Thus, from the view of the company side, the objective is to find a stable matching using the least number of interviews. Our AE arm-proposing DA algorithm is the case that job seekers propose to companies, and companies organize interviews to filter out the best candidates.
>
> Please find responses to specific questions below:
> 1. We use this assumption to stay consistent with previous papers where they assumed $N \leq K$ so that each agent at least can be matched to an arm. They had this assumption so that the regret would not be $\Omega(T)$ for the unmatched agent. Since we do not consider regrets, we agree with the reviewer that such assumption is not required. In fact, removing this assumption does not change our theoretical results, Thm 1 to Thm 5.
>
> 2. Please note that the sample complexity depends on minimum preference gap $\Delta$. In experiments, we keep $\Delta=1$ to focus on comparing algorithm performance rather than the impact of $\Delta$. Thus, each agent’s utilities are permuted from $\{1, 2, \ldots, 20 \}$ following a similar experimental setup in Liu et al. [2021], though with slight differences in permutation.
> Allowing $\Delta$ to vary randomly would obscure algorithm comparisons, as their performance would fluctuate with $\Delta$. For example, uniformly sampling utilities in (0, 20] would result in $\Delta$ ranging from 0 to 20, causing high performance variance.
>
> Other distributions, such as exponentially or polynomially growing, are some potential settings but have flaws especially when the number of agents/arms is large. If utilities are permuted from $1^2, 2^2, 3^2, \ldots, 20^2$, the total number of samples to find a stable matching should decrease, since it should be easier to rank through sampling. E.g. it takes much less effort to differentiate $19^2 = 361$ with $20^2 = 400$ (as compared to differentiate $19$ with $20$).
>
> SPC setting: Preferences where one side has a masterlist capture natural structures, such as riders ranking drivers by ratings or colleges ranking students by exam scores. Previously studies by Sankararaman et al. [2021], Basu et al. [2021], and Wang and Li [2024] have studied this setting. Masterlist is a special case of SPC, both of which ensure a unique stable matching. Thus, we also provide results for SPC to capture this class in our experiments.
> Ref: Wang and Li. Optimal analysis for bandit learning in matching markets with serial dictatorship. TCS 2024
>
> 3. We thank the reviewer for pointing us to the references. Studying approximate stability could be a future research question. Also, it is not clear if the sample complexity would improve asymptotically if we consider approximate stability [2] or [3]. Thus, we first study the sample complexity for stable matchings in our model before exploring approximate stability.
>
> 4. Please note that we do not restrict the value of the utility in our theoretical analysis. Since we consider scenarios (school choice, ridesharing, etc) where we have a finite number of agents and arms, we can assume that the utility values can be scaled as long as it can be stored and accessed using one machine query. In some previous papers that emphasized on regret, e.g. Kong and Li [2024], they consider the case that the utility is a real value in $(0,1]$. Hence, in such an assumption you could expect the minimum preference gap to scale down as the market size increases.
>
> 5. Yes, we agree with the reviewer that studying indifference is an interesting future direction. However, this line of results fails to capture indifference, as the regret bound and sample complexity is proportional to $1/\Delta^2$. One of our initial ideas to overcome the difficulty is to take a different sampling distribution when agents pull an arm, as compared to sub-Gaussian distribution. Indifferences pose several computational challenges (see response to comments).

---

> > ### Comment · Reviewer_wAZr · 2024-08-12
> >
> > I appreciate the very detailed response from the authors. I have taken the discussion into consideration and adjusted my evaluation accordingly to reflect my current judgment:
> > 1. I agree that the incompatibility with indifference in preferences is, unfortunately, common to prior works in this line of literature. I invite the authors to discuss such limitation in the paper and the ideas to overcome it.
> > 2. It seems that the authors are thoughtful with the choice of utility distribution in the evaluation. While I still don't think it is necessarily the most convincing one, the additional argument now makes it more appealing to me. I might suggest including additional discussion on the rational of such a choice, as summarized in the rebuttal comment.
> > 3. Overall, I see the contribution from this work but personally I still think it falls short of the bar for NeurIPS. In a very short summary, I did not find the "at best infinite" argument along with the simulation convincing enough (maybe some average case notion of improvement can be more helpful?), and I still have reservations over the practical significance of the results.

---

> ### Author Response · Authors · 2024-08-12
>
> Thank you for your responses, increasing the score, and taking the time to read our responses.
>
> 1.⁠ ⁠We discuss about ties in limitation.
>
> 2.⁠ ⁠⁠As we mentioned, since our choice of utilities are similar to experiments in previous papers [Liu 2021], we did not discuss this. We can add it briefly in revision.
>
> 3.⁠ Please note in this line of bandit learning in matching markets, the true utility is given and we do not have any assumption about the utility distribution. Thus our use of envy-set (Theorem 5) exploits the structure of the input preference. Since the utilities are part of the input, we cannot discuss about the average case but we provided the best case and the worst case improvements over the uniform DA algorithms in Lemma 3. For an example of practical scenario, please see our labor market example in rebuttal.

---

### Official Review · Reviewer_Uabi · 2024-08-05

**Soundness:** 3
**Presentation:** 3
**Contribution:** 3
**Rating:** 6
**Confidence:** 4

**Summary:**

This paper studies the unexplored question of stability in the study of two-sided market matching problems where preferences of one side are unknown and have to be progressively learned through a bandit learning mechanism. The paper makes a significant contribution in this area, which has hitherto only considered the regret cost aspect. Yet it still neglects the fairness aspect of the ensuing matching, which is a weakness in most solution ignoring balance cost and sex equality cost.

It would be interesting to consider the cost of eliciting preferences and solving the problem in the classic manner, and illustrate the benefit of the proposed solution over that approach.

**Strengths:**

S1. Solid improvement in terms of probability to yield stable solutions leading to Arm Elimination arm-DA algorithm.
S2. Derivation of bounds on samples needed to reach stable matching w.h.p.
S3. Experimental study among proposed variants.

**Weaknesses:**

W1. No consideration of fairness aspects of the ensuing matching.
W2. No illustration of benefit vs. solution that elicits all preferences.

**Questions:**

What is benefit of the proposed methods over one that pays the cost to elicit all preferences?

**Limitations:**

Yes.

---

> ### Author Rebuttal · Authors · 2024-08-06
>
> We thank the reviewer for the interesting questions. Please find responses to specific questions below.
>
> >What is benefit of the proposed methods over one that pays the cost to elicit all preferences?
>
> Response: In the classical preference elicitation framework, each query (either it is based on interview or comparing two options) is deterministic. However, our model enables us to consider stochastic rewards where agents receive a stochastic signal when pulling arms.
> In addition, in a variety of applications (e.g. job interviews), the information is noisy as a candidate’s performance may not be the same in each iteration. Our model enables us to encode such practical scenarios (please also see Line 28-30 in our paper).
>
> > No consideration of fairness aspects of the ensuing matching.
>
> Response: We agree with the reviewer that the fairness aspect of the matching is important. Our algorithms, either aiming to reach an agent-optimal stable solution or arm-optimal stable solution, are good for only one side of the market. Within the matching literature, achieving fairness based on criteria such as sex-equality or maximin fairness either are computationally intractable or are solely defined based on ordinal preferences. In particular, computing a sex-equal stable matching [Kato 1993] or a balance stable matching [Feder 1990] is NP-hard even when all preferences are known and deterministic.
>
> Kato, Akiko. "Complexity of the sex-equal stable marriage problem." Japan Journal of Industrial and Applied Mathematics 10 (1993): 1-19.
>
> Tomás Feder. "Stable networks and product graphs", PhD thesis, Stanford University, 1990.

---

### Decision · Program_Chairs · 2024-09-25

**Decision:**

Accept (poster)

**Comment:**

The review team is relatively positive about the authors' work. On the plus side, reviewers found that the studied problem is novel, considering "two-sided market matching problems where preferences of one side are unknown and have to be progressively learned". The reviewers generally also found the results of the paper interesting and relatively intuitive, but with two caveats:
- One reviewer notes that the analysis is relatively standard and the improvements compared to previous work seem to be a bit weak
- Several reviewers point out issues with the motivation, noting that the problem could be a better-motivated and that there may be a disconnect between the theory/model/setting and the actual intended applications, which is a potential worry.

For this reason, I am recommending acceptance as a poster. My understanding is that the paper is above the bar for NeurIPS if there is space for it, however it still suffers from weaknesses that could be addressed in a re-submission/revision. I set my confidence as "less certain" due to the balance of strengths and weaknesses and I do not feel that I have sufficient information about competing papers to make an informed decision, and further recommend that this paper is discussed with the SAC.